# TenSet: A Large-scale Program Performance Dataset for Learned Tensor Compilers

Lianmin Zheng[1*]  Ruochen Liu[1*]  Junru Shao[2]  Tianqi Chen[23]
Joseph E. Gonzalez[1]  Ion Stoica[1]  Ameer Haj-Ali[1]
[1]UC Berkeley  [2]OctoML  [3]Carnegie Mellon University

## Abstract

Search-based tensor compilers can greatly accelerate the execution of machine learning models by generating high-performance tensor programs, such as matrix multiplications and convolutions. These compilers take a high-level mathematical expression as an input and search for the fastest low-level implementation. At the core of the search procedure is a cost model, which estimates the performance of different implementations to reduce the frequency of time-consuming on-device measurements. There has been a growing interest in using deep learning techniques to learn a cost model to ease the effort of building an analytical model. To realize the potential of such deep learning models, a standard dataset for pre-training and benchmarking learned cost models is necessary. However, this dataset is lacking.

We introduce TenSet, a large-scale tensor program performance dataset. TenSet contains 52 million program performance records collected from 6 hardware platforms. We provide comprehensive studies on how to learn and evaluate the cost models, including data collection, model architectures, loss functions, transfer learning, and evaluation metrics. We also show that a cost model pre-trained on TenSet can accelerate the search time in the state-of-the-art tensor compiler by up to $10\times$. The dataset is available at `https://github.com/tlc-pack/tenset`.

## 1 Introduction

Efficient execution of machine learning models relies on high-performance tensor programs, *i.e.*, optimized low-level implementations of tensor operators such as convolution and matrix multiplication. However, it is notoriously challenging to obtain performant tensor programs for numerous tensor operators on various hardware platforms[7]. Existing systems mainly rely on vendor-provided kernel libraries such as cuDNN [14] and OneDNN [23]. However, crafting these libraries requires spending significant engineering efforts on manual tuning. Moreover, they fall short of supporting new operators invented by researchers and graph optimizations such as operator fusion [12]. To overcome the limitations of manually optimized libraries, researchers and practitioners are building search-based tensor compilers [1, 12, 48]. Given an operator or a computation graph in mathematical expression, these compilers search for the best compiled tensor programs for the target hardware platform. At the core of the search procedure is a cost model, which estimates the performance of tensor program candidates to reduce the time-consuming on-device measurements.

With the advances of deep learning, there has been a growing interest in using deep learning techniques to learn a cost model [5, 13, 20, 25, 39, 48]. By learning directly from the data, learning-based approaches can simplify the development of analytical cost models, especially for complicated modern hardware platforms. However, training data collection is one of the biggest challenges for adopting learning-based approaches in search-based tensor compilers [36]. Currently, the community lacks a public large-scale dataset that can include performance measurements from multiple hardware

---

*Equal contribution.

35th Conference on Neural Information Processing Systems (NeurIPS 2021) Track on Datasets and Benchmarks.

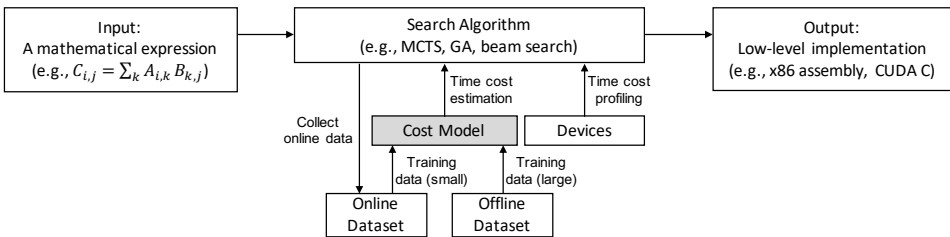

Figure 1: The architecture of a search-based compiler with a learned cost model. The compiler takes a high-level mathematical expression as an input and searches for the best low-level implementation.

platforms. This hinders the development of learning-based approaches as pre-training a decent cost model offline requires a comprehensive dataset. Therefore, some compilers [12, 48] choose to collect the training data online during the search, which makes the search very time-consuming due to unavoidable on-device measurements [27]. Furthermore, without a standard dataset, it is not easy to fairly benchmark and evaluate the proposed models and training algorithms.

We introduce TenSet, a large-scale program performance dataset for learned tensor compilers. TenSet contains 52 million program performance records collected from real measurements on Intel CPUs, AMD CPUs, ARM CPUs, and NVIDIA GPUs. We generate random tensor programs for popular deep learning models. The generated programs are compiled by TVM compiler [12] and measured on the target hardware platforms. With this dataset, we provide comprehensive studies on how to learn and evaluate cost models, including data collection, model architectures, loss functions, transfer learning, and evaluation metrics. We integrate the cost models pre-trained on TenSet into Ansor[48], a state-of-the-art search framework in TVM compiler, and show that it reduces the search time by up to 10× while achieving the same search quality.

## 2   Background: Search-based Compilers with Learned Cost Models

Figure 1 shows the general architecture of a search-based compiler with a learned cost model. This architecture is used by plenty of recent tensor compilers, such as TVM [13, 48], Halide [1, 3, 39], Tiramisu [5] and XLA [25]. The compiler accepts an operator or a computational graph in high-level mathematical expression as input, and runs a search algorithm to find the best tensor program. The adopted search algorithms include Monte Carlo tree search (MCTS)[5, 20], genetic algorithm (GA) [48], beam search[1], simulated annealing [13], and reinforcement learning [2]. The majority of the considered search spaces include loop optimizations such as tiling, vectorization, parallelization, unrolling, and fusion. By searching in a very large space of different optimization combinations, these compilers can often find programs that are better than hand-optimized implementations.

During the search, the algorithm generates a set of promising programs from the search space and compares their performance. The performance can either be estimated by querying the cost model or measured by actually running the programs on the hardware. Due to the size of the search space and the time-consuming on-device measurement, it is impossible to measure the execution time of every program candidate. Therefore, it is common to use a learned cost model to guide the search. The quality of the cost model is one of the most important factors for search efficiency and result quality. To train the cost model, the compilers can use large offline datasets collected in advance, or small online datasets collected on-the-fly during the search, or both.

Many learned cost models have been proposed [1, 3, 5, 13, 25, 39, 48]. They collect their own dataset, use different feature extraction, model architectures, compilers, and hardware platforms. However, they often do not release code or complete datasets that are easy to access and well documented. In addition, they typically only cover one hardware platform, making it hard to use the models in a real multi-backend compiler and hard to study transfer learning across hardware platforms. The fragmented development hinders the research in this area.

Notably, TVM [12] is the state-of-the-art tensor compiler that implements the above search-based architecture. TVM has two generations of search frameworks: AutoTVM [13] and Ansor [48]. AutoTVM is a semi-automated framework, which requires pre-defined manual templates, while Ansor is a more advanced, fully automated framework. This work is built on top of Ansor. Currently, due to the lack of a large-scale offline dataset, TVM has to collect data on-the-fly during the search, leading to an extremely long search time. It can take several hours to optimize and compile a single neural network [13, 48].

```
Data = PLACEHOLDER [1, 56, 56, 64]
PaddedInput(i0, i1, i2, i3) = tir.if_then_else(((((i1 >= 1) && (i1 < 57)) && (i2 >= 1)) && (i2 < 57)),
                                    Data[i0, (i1 - 1), (i2 - 1), i3], 0f)
Weight = PLACEHOLDER [3, 3, 64, 128]
Conv2dOutput(nn, yy, xx, ff) += (PaddedInput[nn, ((yy*2) + ry), ((xx*2) + rx), rc] * Weight[ry, rx, rc, ff])
Bias = PLACEHOLDER [1, 1, 1, 128]
T_add(ax0, ax1, ax2, ax3) = (Conv2dOutput[ax0, ax1, ax2, ax3] + Bias[ax0, 0, 0, ax3])
T_relu(ax0, ax1, ax2, ax3) = max(T_add[ax0, ax1, ax2, ax3], 0f)
```

Figure 2: The computational graph for a fused conv2d-biad_add-relu task.

```
parallel ax0.0@ax1.0@ax2.0@ (0,4)
  for i1 (0,57)
    for i2 ((floormod(ax0.outer.outer...
      for i3 (0,64)
        PaddedInput = ...
  for ax3.0 (0,2)
    for ax2.1 (0,7)
      for ax3.1 (0,8)
        Conv2dOutput auto_unroll: 16
        for rx.0 (0,3)
          for rc.0 (0,4)
            for ry.1 (0,3)
              for rc.1 (0,16)
                for yy.3 (0,28)
                  vectorize ff.3 (0,8)
                    Conv2dOutput = ...
    for ax1.2 (0,28)
      vectorize ax3.2 (0,8)
        T_relu = ...
```

| Item | Number |
|------|--------|
| Networks | 120 |
| Hardware Platforms | 6 |
| Tasks | 13,848 |
| Measurement records | 51,532,994 |

Figure 3: A sample program for the task in Fig. 2      Table 1: Dataset statistics

# 3 TenSet: A Dataset for Tensor Programs

A good dataset is the first requirement of a good model [42, 36]. The purpose of TenSet is to provide a large-scale dataset for pre-training and benchmarking the cost models in tensor compilers. This section describes the requirements of a good dataset and introduces the contents of TenSet.

## 3.1 Requirements of the Dataset

**Large-scale.** A learned cost model using this dataset is expected to perform well on common workloads and generalize relatively well to other uncommon workloads. A large-scale dataset containing diverse workloads is necessary for generalization ability.

**Multi-platform.** A tensor compiler typically supports multiple hardware platforms. The dataset should thus contain the records from multiple platforms. This can be used to train different models for different platforms. If the dataset also contains the performance of the same program on different platforms, it is possible to let the model learn the difference among different hardware platforms. This enables advanced research on transfer learning among different hardware platforms.

To the best of our knowledge, TenSet is the first public tensor program dataset that meets these two requirements. With the death of Moore's law, we are seeing a lot of new custom hardware (continuously changing hardware) and this dataset should generalize well to new programs and new hardware.

## 3.2 Terminology

We define some terminologies used in the rest of this section.

**Network and Subgraph**: A deep neural network with a specific input shape. The input shape contains the batch size and image size (or sequence length). A network is a computational graph. Typically, tensor compilers partition a large computational graph into several small subgraphs based on certain rules. Subgraphs are the finest granularity for compilation.

**Hardware platform**: A device to execute the subgraphs. Note that devices with different hardware architectures (e.g., NVIDIA Tesla and Volta) are counted as different hardware platforms.

**Task**: A task is a pair of a subgraph and a hardware platform. A network contains many subgraphs, so we can extract many tasks from a network on a hardware platform. For example, after the graph partitioning for ResNet-50, there are 27 unique subgraphs, which implies 27 search tasks.

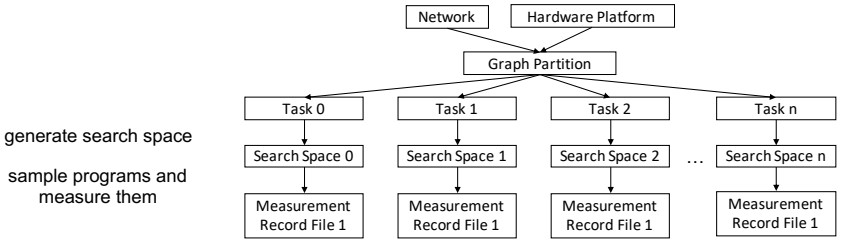

Figure 4: The hierarchical structure of the dataset.

| Hardware Platform | Cloud Instance | Other Comments |
|---|---|---|
| Intel Platinum 8272CL @ 2.60GHz (16 cores) | Azure D32s_v4 | AVX-512 |
| Intel E5-2673 v4 @ 2.30GHz (8 cores) | Azure F16s | AVX-2 |
| AMD EPYC 7452 @ 2.35GHz (4 cores) | Azure D16as_v4 | AVX-2 |
| ARM Graviton2 (16 cores) | AWS c6g.4xlarge | Neon |
| NVIDIA Tesla K80 | AWS p2.xlarge | Kepler Architecture |
| NVIDIA Tesla T4 | AWS g4dn.xlarge | Turing Architecture |

Table 2: Hardware Platforms in this Dataset

**Search space**: Each task has its own search space, which is determined by the input/output tensor shapes, data types, data layouts, and the target hardware platform. The search space is usually in the order of millions on CPUs and billions on GPUs.

**Program**: A program or a tensor program refers to a low-level, hardware-dependent implementation of a subgraph. It can be seen as a candidate in the search space.

**Measurement record**: A measurement record is a tuple of task, program, and on-device measurement result. The program measurement module takes the task and program, performs the compilation, execution, and measurement. The measurement result contains the execution time of the program or an error code if encountering compilation or runtime errors.

## 3.3 An Illustrative Example of Tensor Programs

Figure 2 shows the computational graph of an example task, which is a fused convolution + bias add + ReLU activation. The computational graph is printed in a form similar to Einstein notation. Figure 3 then shows a sample program from the search space of this task on a CPU. The program is optimized by multi-level tiling, parallelization, vectorization, unrolling, and fusion.

## 3.4 Contents and Data Collection

The dataset is organized in a hierarchical structure, as shown in Figure 4. The top-level includes some common networks with different configurations and hardware platforms. Table 6 (in Appendix A) lists the specifications of network architectures and input shapes. The network architectures are chosen from PyTorch's vision model Zoo and Huggingface's transformer model Zoo. They cover both representative CV and NLP tasks. We vary the batch size and input image sizes to generate different subgraphs. Note that we focus on small batch sizes in this dataset because tensor compilers are mainly used for optimizing trained models for inference. In total, there are 120 network configurations.

Next, we pick 6 hardware platforms according to their availability on public cloud providers, their popularity in the machine learning community, their compiler toolchain support, and the data collation cost. Their specification is listed in Table 2. We favor publicly-accessible cloud instances for improved reproducibility.

For each pair of (network, hardware platform), we run the graph partitioning algorithm to obtain a list of unique subgraphs. A subgraph usually includes a heavy tensor operator (e.g., conv2d, conv3d, conv2d_tranpose, depthwise_conv2d, matmul, softmax) fused with lightweight operators (e.g, element-wise operators). For each subgraph, we randomly sample programs from its search space generated by Ansor [48]. We then dump the sampled programs and measure them on AWS and Azure cloud instances. For each program, we do warm-up and run several repeated measurements.

The time costs of all repeated measurements are saved in the measurement record files. The record file is in JSON format. Read and write utility functions are provided to parse and generate these files.

Figure 1 shows some statistics of the dataset. In total, we collect tasks from 120 networks × 6 hardware platforms. There are 2,308 subgraphs extracted from the 120 networks, so we have 2,308×6=13,848 tasks for 6 platforms in total. For each task, we sample at most 4,000 programs from its search space and generate measurement records. The search space size of a task varies from 10 to the order of billions. In total, the dataset includes 51,577,248 measurement records from all the tasks. The collection process takes several weeks with clusters of cloud instances.

## 4 Learning and Evaluating a Cost Model in Tensor Compilers

The goal of a learned cost model is to rank the performance of different tensor programs in a given search space. This section gives a high-level overview of how to learn, evaluate and use cost models in tensor compilers. The scope of this work is supervised learning.

### 4.1 Learning a Cost Model

**Feature Extraction** To feed a program into a machine learning model, the compiler can extract features from the high-level task description, optimization specification, and low-level compiled program. Currently, manual feature extraction is still required because the programs contain a lot of structural and numerical information. Learning an end-to-end model directly from text tokens has not been explored for this specific problem in the literature. The features can be extracted at multiple levels, from high-level task descriptions to low-level compiled programs. Features from higher levels are faster to extract, while features from lower levels are slower to extract because they require going through the compilation process. The high-level task features can include the features or embeddings of the input computational graphs such as shape and access pattern. It can also include hardware platform information such as cache size and vector width. The optimization features can include the used loop transformations and schedules. The low-level program features can include features extracted from the lowered IR or even machine code, which can help to model the end-to-end compilation process. At all levels, the features are used to capture the memory access and computation patterns of the tensor programs, which are the most important factors of the performance of the programs. The lists of typical features can be found at [1, 5, 48]. The specific features we use can be found in Appendix C.

**Model Architecture** The features that can be extracted can have vector structures, tree structures, or graph structures. They also have variable lengths. To feed these features into a machine learning model, the hierarchical structure is generally flattened and fed into feed-forward neural networks with padding or sum aggregation. The hierarchical structure can also be feed directly into tree recurrent neural networks and graph neural networks. Existing works have adopted MLP [1], LSTM [5, 39], GRU [13], GraphSAGE [25, 22], GCN [35], and GBDT [48, 11].

**Loss Function** The model is used to rank the performance of candidates in a search space. Therefore, the model can be trained with regression losses to predict absolute scores or be trained with ranking losses to predict relative scores. For regression losses, models can be trained with Mean Square Error (MSE) loss to predict the normalized throughput or latency of a program [1]. For ranking losses, models in previous works have been trained with pairwise or listwise losses [10, 13].

**Transfer Learning** During the search of a new task, if new online data can be collected from the task, it can be used to adapt the pre-trained model to the new task with transfer learning. The transfer learning can be done by using transferable features, fine-tuning, learning local models [13], or meta-learning [35, 18].

### 4.2 Integration with the Search Algorithm

To use a cost model in the search process, the compiler runs a search algorithm and picks the top-$k$ programs according to the cost model for each task. If on-device measurements are not allowed, the $k$ is set to 1 and the compiler makes decisions totally based on the cost model. If on-device measurements are allowed, the $k$ can be set to a larger number. For each task, the compiler measures the top-$k$ programs on actual devices and picks the best one according to real measurement results.

|         | RMSE | $R^2$ | Pairwise Accuracy | Top-1 Score | Top-5 Score | Latency (ms) |
|---------|------|-------|-------------------|-------------|-------------|--------------|
| Model #1 | 0.09 | 0.77 | 0.85 | 0.86 | 0.92 | 7.89 |
| Model #2 | 0.07 | 0.89 | 0.84 | 0.87 | 0.95 | 6.45 |
| Model #3 | 7.27 | -1818.41 [2] | 0.89 | 0.88 | 0.96 | 6.39 |

Table 3: Evaluation of different models using dataset-based metrics and search-based metrics. The description of each metric is detailed in Appendix B. Lower RMSE and latency, and higher pairwise accuracy, $R^2$, and top-$k$ scores are desirable.

This process can also be done iteratively because we can update the cost model using the newly collected measurement data and run the search again. We can also use a scheduler to allocate different time budgets to different tasks according to their importance [47, 48]. In a search with on-device measurements, a number of total measurement trials is set as the time budget, because measurements are the most expensive part of the search. The search terminates when it runs out of allowed measurement trials.

### 4.3 Evaluation Metrics

To evaluate and compare the performance of cost models, there are two types of evaluation metrics: *dataset-based metrics* and *search-based metrics*. The dataset-based metrics evaluate the accuracy of the model on a static dataset, and search-based metrics evaluate the end-to-end search efficiency or search quality after integrating the cost models into the search algorithms.

Typical dataset-based metrics include Rooted Mean Square Error (RMSE), Mean Absolute Percentage Error (MAPE), $R^2$ (Coefficient of Determination), pairwise comparison accuracy. We also propose a new metric top-$k$ score, which reflects how well the top-$k$ programs predicted by the model perform. Their definition can be found at Appendix B. For search-based metrics, there are two typical metrics: we can either fix the search time, and compare the latency of the resulting programs, or fix a converged latency, and compare the search time used to reach it.

The search-based metrics are the end-to-end objective of a search-based compiler, but these metrics are expensive to compute due to the time-consuming search process. These metrics also involve other factors that are not directly related to the cost model. On the other hand, dataset-based metrics can be computed very fast on a static dataset. They are more directly related to the cost model but do not directly reflect the end-to-end objective. In Sec. 5.1, we compare several dataset-based metrics to see how well they reflect the end-to-end objective.

## 5 Experiments

In this section, we try to answer the following questions: What are the best metrics to evaluate a cost model (Sec. 5.1)? How do the model architectures and loss functions influence the model performance (Sec. 5.2)? How should we collect the dataset (Sec. 5.3)? How can the model improve search efficiency (Sec. 5.4)? Is the online transfer learning useful (Sec. 5.5)? How can we further improve the search results (Sec .5.6)?

### 5.1 Evaluation Metrics

In Sec. 4.3, we list some widely used dataset-based metrics and discuss the possible discrepancies between these metrics and the end-to-end objective. In this section, we compare several dataset-based metrics and pick the one that reflects the end-to-end objective best. We can then use it as the evaluation metric in the following sections without involving the search process. Additional experimental setup can be found in Appendix C.

We train three different models and evaluate them with both dataset-based metrics and search-based metrics. We use the dataset from Intel Xeon Platinum-8272. We hold out a ResNet-50 (batch size=1, image size=224) as the test set, and use the rest of the dataset as the training set. For dataset-based metrics, we evaluate the models on the test dataset. For the search-based metrics, we run the search

---

[2]A negative $R^2$ is possible when the model performs worse than the baseline model which always predicts the mean value. This means the model is not trained to be a good regression model.

|  |  | ResNet-50 | MobileNet-V2 | ResNext-50 | BERT-tiny | BERT-base |
|---|---|---|---|---|---|---|
| MLP + Ranking loss | Top-1 Score | 0.8823 | **0.7446** | **0.8584** | **0.8041** | **0.9143** |
|  | Top-5 Score | **0.9456** | 0.9027 | **0.9502** | 0.8601 | **0.9753** |
| MLP + MSE | Top-1 Score | **0.8873** | 0.7026 | 0.6772 | 0.8001 | 0.8535 |
|  | Top-5 Score | 0.9371 | 0.8844 | 0.9167 | **0.8805** | 0.9387 |
| XGBoost + MSE | Top-1 Score | 0.8535 | 0.7259 | 0.8411 | 0.6534 | 0.7621 |
|  | Top-5 Score | 0.9341 | **0.9085** | 0.8958 | 0.8522 | 0.9441 |
| LSTM + MSE | Top-1 Score | 0.8637 | 0.7145 | 0.7653 | 0.7972 | 0.8693 |
|  | Top-5 Score | 0.9239 | 0.8842 | 0.9173 | 0.8295 | 0.9354 |

Table 4: The top-$k$ scores of different cost models on five test networks. A higher score is better. Scores in bold are the highest top-$k$ score for each network.

algorithm with each model for ResNet-50 under the same time budget, and measure the latency of the result programs. Table 3 shows the evaluation results of five different dataset-based metrics and a search-based metric. For dataset-based metrics, a good model has the RMSE close to 0 and the pairwise accuracy, $R^2$ and top-$k$ score close to 1. For search-based metrics, the lower latency, the better the search result is.

According to Table 3, although Model #1 has a small RMSE and large $R^2$, it does not result in as good latency as Model #3 which has a high RMSE and an extremely low $R^2$. This suggests that RMSE and $R^2$ might not be the appropriate metrics for models trained with ranking losses. Model #3 is trained with ranking loss. Directly using its output values to compute regression metrics such as RMSE and $R^2$ gives meaningless values. On the other hand, pairwise comparison accuracy and the top-$k$ score are more consistent with the final latency; thus they are better dataset-based metrics for our use case. One slight difference between these two is that pairwise comparison accuracy does not differentiate Model #1 and Model #2 well, so top-$k$ score reflects the end-to-end objective best. Intuitively, the problem we are tackling is fundamentally a ranking problem rather than a regression problem. Therefore, using ranking-based metrics can evaluate all kinds of models while directly using regression-based metrics is meaningless for some models. In addition, we care the most about the top-ranked programs because the compiler will only choose and compile them. We use top-$k$ score in the following sections.

## 5.2 Model Architectures and Loss Functions

This subsection evaluates the effectiveness of different model architectures and loss functions. We use the dataset from Intel Xeon Platinum-8272. We hold out a test set that consists of five networks, ResNet-50, MobileNet-V2, ResNext-50. BERT-tiny, and BERT-base with batch size 1 and image size 224 (or sequence length 128), and train models using different combinations of architectures and loss functions on the training set that consists of the whole dataset excluding all the tasks that appear in the above five networks. Further training details are included in Appendix C. Note that for the ranking loss, we use the probabilistic cost function originated from LambdaRank [44]. Table. 4 shows the top-$k$ scores of these models on the test set. The highest top-1 score and top-5 score in each column are in bold. In general, a MLP model trained with the ranking loss performs the best.

## 5.3 Dataset Size

This section evaluates the impact of dataset size on model performance. The size of a dataset is defined as # tasks multiplied by # programs per task. In dataset collection, we typically have a budget on the total size, but how to set these two factors given the total size (i.e., their product) is unclear. To give guidance to future dataset collection, we compare the performance of cost models trained with different combinations of # task and # programs per task. We use the same training set and test set as in Sec. 5.2 but randomly downsample the whole training set to get smaller datasets with different combinations of # task and # programs per task. Figure 5a and Figure 5b show how the model's top-$k$ score varies along with changes in the two factors described above. It shows that the model's performance generally increases along with the dataset size, although an increase in the dataset size does not guarantee an improvement in the model performance. Besides, the figure shows that in general, given a fixed total size, allocating more to # task is often more effective than allocating more

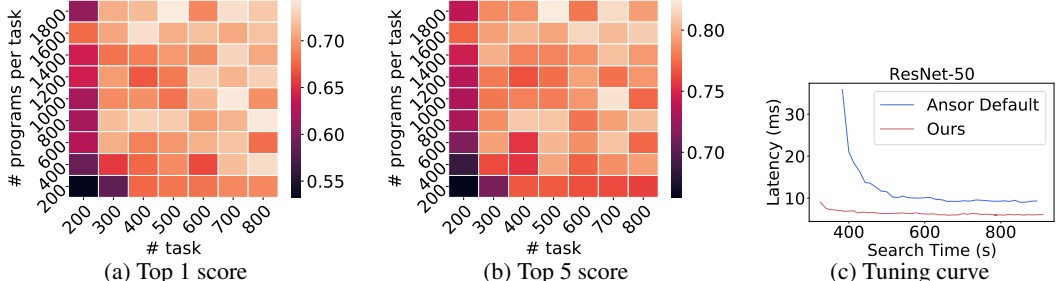

(a) Top 1 score        (b) Top 5 score        (c) Tuning curve

Figure 5: (a)(b): The impact of dataset size on model performance. The x-axis is number of tasks and y-axis is number of programs per task. The lighter the color, the better the model. (c): Network performance tuning curve. The y-axis is the result program's latency and the x-axis is the search time.

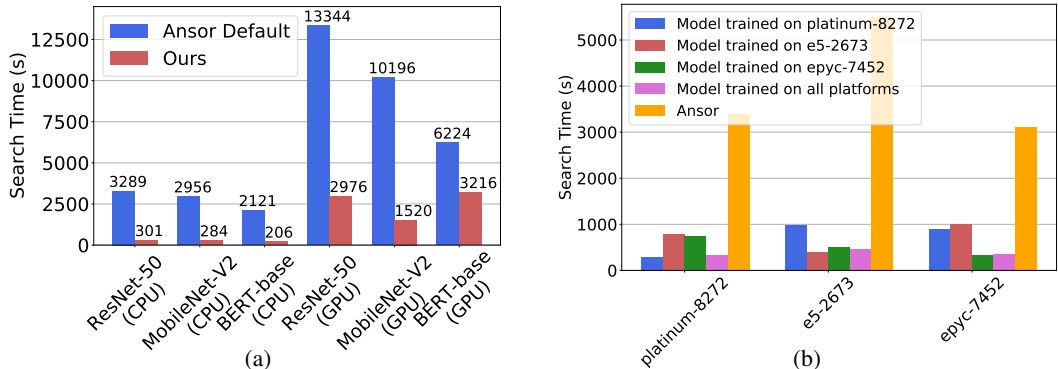

(a)                       (b)

Figure 6: Search time comparison. The y-axis is the search time used to converge to the same result. (a) Shows the search time for different neural network architectures on a CPU and a GPU. (b) shows the search time on different hardware platforms for models trained on different hardware platforms.

to # program per task. For example, for a dataset of size 120000, (600 tasks, 200 programs per task) works better than (300 tasks, 400 programs per task) and (200 tasks, 600 programs per task).

## 5.4 Search with a Pre-trained Cost Model

We combine the findings from the previous sections and train four MLP models with ranking loss for Intel Xeon Platinum-8272, Intel E5-2673, AMD EPYC-7452, and NVIDIA K80 respectively. Each model is trained on the dataset collected from a single hardware platform. We integrate the cost models into the Ansor auto-scheduler[48] in TVM [13]. We name the original Ansor (i.e., the one without our cost model) as "Ansor default" and follow its official benchmark scripts to run the benchmark.

In Figure 5c, we run the search for ResNet-50 on Intel Xeon Platinum-8272 with the cost model trained on the same hardware platform, and report how the result program's latency changes along with the total search time in both Ansor default and our approach. It shows that our pre-trained cost model makes the search converge much faster. This is because "Ansor default" does not have a pre-trained model. It thus has to start from random search and collect data during the search and train the model online, which takes an extremely long time. Figure 6a compares the search time that Ansor default and our approach take respectively to converge to the same result on more networks and two hardware platforms (Intel Platinum CPU and NVIDIA K80 GPU). It shows that the use of our cost model reduces the search time by up to $10\times$ while maintaining the same search quality. One of our industry collaborators runs weekly benchmarks and compiles hundreds of models for dozens of hardware platforms. The long-term savings of using this dataset are significant.

In Figure 6b, we run the search algorithm with three cost models on three different hardware platforms. It shows that the models are able to transfer across different hardware platforms, although the performance is not as good as on the platform that generates the dataset that the model is trained

|  |  | Intel Xeon Platinum-8272 | Intel e5-2673 |
|---|---|---|---|
| Setting 1 | With transfer learning | 6.22 ms | 27.26 ms |
|  | Without transfer learning | 6.43 ms | 29.94 ms |
| Setting 2 | With transfer learning | 6.44 ms | 28.92 ms |
|  | Without transfer learning | 7.15 ms | 32.03 ms |

Table 5: Evaluation of the effectiveness of transfer learning. The model is trained on Intel Xeon Platinum-8272 and evaluated on both Intel Xeon Platinum-8272 and Intel e5-2673. We compare the latency (ms) of the result program with a fixed number of measurement trials on ResNet-50.

on. We also train a model on a training set that consists of one-third of the data from each platform. This model performs the second best on all three platforms, which implies the possibility to train one general model for all hardware platforms.

## 5.5 Transfer Learning

To explore transfer learning, the cost model is pre-trained using offline learning with a static dataset and later improved using online learning during the search. More specifically, we first collect a certain number of measurement records for each task during the search, and fit a local model to predict the difference between the measured latency and the pre-trained model's prediction. Then we continue to conduct another round of search, during which we tune the pre-trained model's prediction with the local model's predicted difference. In this experiment, we train a model using the dataset collected from Intel Xeon Platinum-8272 excluding ResNet-50 (batch size 1, image size 224, 240, 256), then run the search algorithm on both Intel Xeon Platinum-8272 and Intel e5-2673. We evaluate the effectiveness of the local model in two ways.

In Setting 1, we report two results: 1) without using transfer learning where we run the search for 50 measurement trials per task and choose the best measurement out of 50 trials and 2) with transfer learning where we use the first 40 out of 50 measurements to fit a local model (the last 10 measurements do not update the model) and choose the best measurement out of 50 trials. In Setting 2, we do everything similar to Setting 1 but we only consider the last 10 measurements per task in the final reported result. The last 10 measurements are programs collected after training the local model, so we can study how exactly the local model affects the programs collected within the same number of trials. Setting 1 is more like an end-to-end benchmark while Setting 2 is more like a micro-benchmark for the transferred model. In both cases, we compare the latency of the resulting program, as shown in Table 5. In both settings, transfer learning improves the search results by producing programs with lower latency.

Note that we only do transfer learning across different types of CPUs. We leave the transfer learning from CPUs to GPUs to future work. Our first intuition is that CPUs and GPUs are based on very different architectures in terms of both memory hierarchy and execution model, and hence we do not expect transfer learning between them would be very useful.

## 5.6 Optimizing with Additional Random Sampling

In Sec. 5.4, when compiling a single network with many tasks, the best program was selected for each intermediate task. For example, for every intermediate neural network layer, the algorithm picks the best performing configuration for that particular layer and combines it with the best performing configuration of the next layer and so on. However, sometimes the combination of the best candidates of different tasks does not result in the best end-to-end performance. In this experiment, we integrate random sampling to solve this problem. After measuring the program composed of best candidates of all tasks, we randomly sample one of the top 3 candidates for every task and repeat this procedure 80 times; for each time, we measure the end-to-end performance of the new program and choose the best performing program so far.

Figure 7 shows the experiment results for 3 networks. For each network, we first run the search process for a certain number of trials and start the random sampling based on the measurement records collected during the search. We then plot the best inference latency we have found as a function of the number of random samples we have explored. The plot shows that as the number of random samples increases, we can always find better programs with lower latency. The measurement

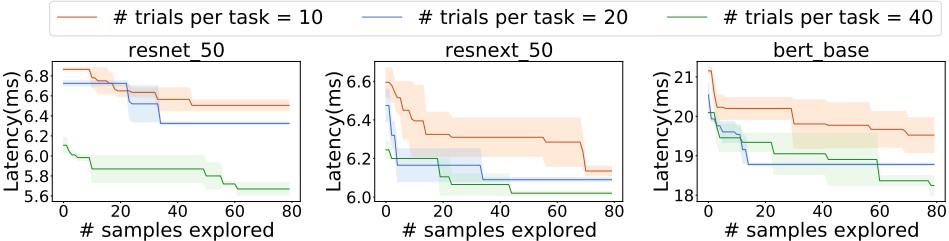

Figure 7: Optimizing with additional random sampling. The figure shows the lowest program latency found so far as a function of the number of times we reran the experiment.

variance is around 0.1 ms, so we can conclude that the decrease in latency is due to improvement in programs instead of noise. Note that although the random sampling generally produces better programs, this process itself takes time, which leads to a trade-off that we have to consider.

## 6    Related Work

Tensor compilers use compiler techniques to optimize the execution of tensor programs. Some notable compilers are Halide [33], TACO [26], XLA [41], Tensor Comprehensions [43], TVM [12], nGraph [16], Glow [34], Tiramisu [6], TASO [24], HummingBird [32], and Rammer [29].

To guide the search in search-based tensor compilers, many learned cost models have been proposed [1, 3, 5, 13, 25, 39, 48]. Besides learned cost models, there are analytical models [8, 28, 38, 40, 43] for tensor program cost estimation. Apart from single device tensor program domain, there are learned cost models for other system problems [30, 46, 37]. In addition to supervised learning approaches, there are reinforcement learning approaches for compiler optimizations [2, 4, 15, 19, 21, 31, 47, 49].

There are several existing performance datasets for programs [6, 17, 45]. To the best of our knowledge, TenSet is the first public multi-platform dataset for tensor programs with the largest number of samples.

## 7    Discussion

**Limitation** There are some limitations of this dataset. For example, the dataset only includes programs for floating point and dense neural networks. The subgraphs are partitioned by a specific algorithm, which limits the types of subgraphs.

**Potential societal impact** The dataset is of tensor programs. It does not contain any personally identifiable information or offensive content. The dataset is used to make tensor compilers better, which makes the execution of neural networks faster. It does not have any direct potential negative societal impact.

**Conclusion** We introduce TenSet, a large-scale multi-platform program performance dataset for learned tensor compilers. We conduct comprehensive experiments with the dataset and show its practical usage in Ansor, the state-of-the-art tensor compiler. We hope that despite the end of Moore's law, and despite the continuously changing application-specific hardware platforms as a result of it, TenSet can help continue the performance scaling by improving the tensor compilers and further advancing the research in the field.

## 8    Acknowledgement

We would like to thank Cody Hao Yu, Zhao Wu, Chengfan Jia, Minmin Sun, Wanchen Sui, Jun Yang, and anonymous reviewers for their insightful feedback. In addition to NSF CISE Expeditions Award CCF-1730628, this research is supported by gifts from Alibaba Group, Amazon Web Services, Ant Group, CapitalOne, Ericsson, Facebook, Futurewei, Google, Intel, Microsoft, Nvidia, Scotiabank, Splunk, and VMware.

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
