# OpenReview forum: "TenSet: A Large-scale Program Performance Dataset for Learned Tensor Compilers"
_NeurIPS.cc/2021/Track/Datasets_and_Benchmarks/Round1 — NeurIPS 2021 Datasets and Benchmarks Track (Round 1)_

### Official Review · Reviewer_rjry · 2021-07-02
**nice dataset of tensor program performance**

**Rating:** 8
**Confidence:** 4
**Clarity:** yes.

**Strengths:**

Nice, well-described and useful dataset.

**Weaknesses:**

Probably somewhat niche area.

**Additional Feedback:**

N/A

**Correctness:**

I'm not an expert enough to comment on the correctness of the data collection, but the description is sound and things appear quite reproducible.

**Documentation:**

Yes, well-documented.

**Ethics:**

No concerns.

**Relation To Prior Work:**

I think well-discussed.

**Summary And Contributions:**

This work presents a dataset of tensor program performance, TenSet. The main differentiation wrt to what's out there already is the fact that the dataset is large-scale and multi-platform. The dataset collection procedure is sound.  I am a bit worried if they networks chosen are representative enough of potential other networks that one could design.

Sections 4 and 5 are very well done, I enjoyed the discussion about Table 3. The transfer learning experiments are certainly exciting and novel.

All in all, this is a great piece of work. I don't believe that it will have a very *broad* appeal (in the same way that a new ImageNet would), but within the community of folks that design such systems, I think a dataset like this will be very useful and a good way to publish stuff for anyone doing e.g. new ML-focused compilers.

---

### Official Review · Reviewer_QK6T · 2021-07-03
**An open dataset to accelerate training models for tensor compilers**

**Rating:** 7
**Confidence:** 3

**Strengths:**

The paper clearly presents how a model-driven tensor compiler works and the type of dataset which can be used to initially train the model for such a compiler. In machine learning, good problem definitions are very valuable.

The experimental insights are useful to anyone looking to construct such a tensor compiler dataset, or make effective use of an existing one.

Generally, the paper is lean and well structured.



**Weaknesses:**

My primary concern is that the paper analyzes a limited range of architectures, and many of its conclusions are about applying the dataset to train models for other architectures within that limited range.  ML accelerators are rapidly diversifying

My biggest question: if you wanted to train an ML model for a significantly different architecture, how much would the utility degrade? I'd like to see a the transfer learning section expanded to consider a subset of the dataset trained on a CPU *only* used to initialize a model for use with a CPU contrasted with the same data used to initialize a model for a GPU (and ideally a TPU). How much better is this than just scratch training the GPU/TPU model?

I'd also like to an analysis of the systemic savings for using a dataset like this one as a starting point relative to just training an architectural specific model from scratch, amortized across the probable lifetime of a ML-model driven tensor compiler. Is this dataset useful for optimizing something that matters?

**Additional Feedback:**

Interesting paper; enjoyed reading it!

**Clarity:**

The paper is clearly written. Two minor nits:

Model is internally overloaded between compiler models and models being compiled. It would help to adopt a consistent modifier for one class.

The use of the term "program" to refer to a specific implementation of a subgraph is a bit confusing, and "implementation" might be better.

**Correctness:**

The dataset construction seems methodologically sound.  This type of dataset has the nice property that ground truth is directly measurable.

**Documentation:**

The documentation is clear and easy to follow.

**Ethics:**

The nature of the data, essentially measurements of system performance by the authors, makes the common ML ethical concerns around privacy and ownership not applicable. Reduction in the compute cost of ML is a societally positive.

**Relation To Prior Work:**

The paper's related work section is comprehensive, but very terse. The paper's primary focus on the dataset and not the compiler or model is unique, but some analysis of how the related work compares with the models and metrics analyzed in the paper would be appreciated.

**Summary And Contributions:**

The paper describes TenSet, a dataset composed of the various ML model subgraphs, each with multiple architecture-specific implementations, and the performance of those implementations on the architectures in question. The dataset is intended for training models used to select implementation for ML subgraphs in ML-driven compilers. The paper first describes how these ML-driven compilers function. It then introduces requirements, terminology used by, contents of, and data collection methodology for the dataset.  It describes how a compiler trains and evaluates the results of a cost model when searching the implementation space. It presents and analyzes experiments on selecting the best evaluation metric, selecting the best model, choosing the the data set size and composition, using pre-trained models, using transfer learning, and optimizing with random sampling. It closes with a brief but dense related work summary and a few words on limitations and societal impact.

The key contributions of the paper, beyond the dataset itself, are the conclusions of the experiments: MLP+ranking loss is a strong model/metric combination, a fixed size dataset is better off with more subgraphs than implementations/programs, transfer learning improves model training time and accuracy, and that random sampling can improve overall graph solution.

---

### Official Review · Reviewer_7d41 · 2021-07-06
**Promising data-driven approach to compiler optimizations**

**Rating:** 7
**Confidence:** 4

**Strengths:**

The authors collect a lot of program, performance pairs, which is surely a large cost. They compare favorably to Ansor, a competitor baseline, both in terms of search speed and ultimate performance, though I am not expert enough to know whether Ansor is state of the art or a tough baseline to beat.

Even when evaluating their cost function on different hardware platforms than they were trained on, their model outperforms Ansor. If Ansor is a strong baseline then this is a good sign for the longevity of this benchmark being useful as new hardware is released which must be optimized, but which is obviously not present in this dataset.

**Weaknesses:**

The hardware dependence and lack of success in transfer learning is a weakness. In table 5 they did show some improvement with transfer learning, but the discrepancy between the Xeon and e5-2673 is still quite large at an absolute ratio of about 4x. This discrepancy is likely to be exacerbated with time as new hardware is released, and so it is unclear for how long this data will be practically useful. On the other hand though, they do explore how much data would be needed to get good results at lower costs, and so perhaps collecting new data for new hardware will become as common as benchmarking. I would like to see more discussions on hardware-specific differences and the effect on generalization, especially between CPU and GPU.

I also find the motivation for the metrics they use to judge the cost functions to be lacking. In other words, I am not simply convinced by only 3 models that, for example R^2 and RMSE are not useful. More on this in the Correctness section.

**Additional Feedback:**

I think this work is very interesting. Overall I recommend the authors do a copy-edit pass, add details in explaining why they made the choices they made (see my other feedback for specifics), and add log-likelihood estimates to their models.

**Clarity:**

The paper has some grammatical errors, but these did not interfere with my understanding. I do think the text should introduce the background context more thoroughly. In particular I would like to understand how strong of a baseline Ansor is (does it generally beat everything else?). I would also like a bit more discussion on some of the arbitrary choices made. For example: why focus on those 6 hardware choices (Why only Nvidia GPUs)? Are random tensor programs representative of realistic tensor programs (what is the variance, min, and max latency of these programs?)? Why did they choose the 120 networks as tasks? Did these cover both NLP and vision and time-series models?

**Correctness:**

The R^2 value for Model #3 in Table 3 caught my eye. I had only thought R^2 should lie between 0 and 1, but I looked it up and apparently there are cases in which it can become strongly negative as presented by the authors. In that case the suggestion appears to be that some poor assumptions were made or that value itself is not actually at all meaningful. I know the authors come to this conclusion, but instead of simply dismissing regression metrics I think this could use some more attention.  I think since this is a more or less standard regression problem, the authors should report log-likelihoods under some assumptions of the latency precision. I suspect a more common log-likelihood will much more closely align with latency.

**Documentation:**

The authors provide a GitHub URL of the code and data. I have not reviewed this, but I assume it's fairly straightforward to consumed and develop their work. They computed everything on standard Amazon Web Service compute nodes, and so (as long as those nodes remain available for use) their work is reproducible. The authors should specify how many cloud instances they used over 'several weeks' in order for others to judge reproduction properly.

**Ethics:**

I am not aware of any ethical problems with this work.

**Relation To Prior Work:**

As I stated before, the comparison to Ansor and others should be explain in better detail. Another issue with Ansor is they compare to the 'default' behavior of Ansor which does not have a 'pre-trained model'. Is this the way the community actually uses Ansor? Is it a fair comparison? Is it easy to modify the Ansor default behavior slightly to get much better results? I don't think authors of new methods are responsible for casting their competitors in absolutely the best light possible, but I think a small amount of discussion here for context will make it much easier to judge the importance of this work.

**Summary And Contributions:**

The authors have collected 52 million pairs of low-level program implementations and their performance on 6 hardware platforms for 120 neural networks. They then use this data to train a cost estimate model which can avoid the expense of literal evaluation of compiled code to improve the speed of compiler optimization searches. They explore which metrics of evaluating cost model candidates most closely correspond with ultimate performance of a given program. They also explore generalization of using training data for one hardware platform to estimate a cost function for another hardware platform, and explore transfer learning to improve generalization.

---

### Author Response · Authors · 2021-07-13
**Revised manuscript following reviewers' comments**

We thank the reviewers for their valuable feedback. Following their comments, we have updated the manuscript with the following modifications:


1. In Sec 3.4, mention why we choose the 120 networks and 6 hardware platforms.
2. In Sec 5.1, clarify why there is a negative R^2 value and the correctness of evaluation metrics.
3. In Sec.5.4, mention that Ansor-default is a strong baseline and we follow its official benchmark script to get the best result out of it.
4. In Sec 5.4, mention the system systemic saving of using the dataset.
5. In Sec 5.5, add our thoughts on transfer learning across different types of hardware platforms.
6. In Sec 6, emphasize how our dataset is different from other existing program performance datasets.
7. Update the text to use “model” for compiler cost models and “network” for the models being compiled.

---

### Decision · Program_Chairs · 2021-07-26

**Decision:**

Accept

**Comment:**

The paper introduces a dataset of tensor program performance records for six commonly used hardware platforms which can be used to facilitate the development of tensor compilers. All reviewers were positive both on the dataset itself, and on the clarity of the paper. Reviewers pointed out some minor details to be clarified which were addressed by the authors’ response and revised paper, and in the end all reviewers agreed that the paper should be accepted.  Congratulations on having your paper accepted to the NeurIPS 2021 Track on Datasets and Benchmarks! The authors are encouraged to take the feedback from reviewers into account when preparing the final version of their paper.